# Ambient Air Pollution and Cardiorespiratory Outcomes amongst Adults Residing in Four Informal Settlements in the Western Province of South Africa

**DOI:** 10.3390/ijerph182413306

**Published:** 2021-12-17

**Authors:** Herman Bagula, Toyib Olaniyan, Kees de Hoogh, Apolline Saucy, Bhawoodien Parker, Joy Leaner, Martin Röösli, Mohamed Aqiel Dalvie

**Affiliations:** 1Centre for Environmental and Occupational Health Research, School of Public Health and Family Medicine, University of Cape Town, Anzio Road, Observatory, Cape Town 7925, South Africa; herman.bagula@gmail.com (H.B.); olaniyanolan@gmail.com (T.O.); 2Department of Epidemiology and Public Health, Swiss Tropical and Public Health Institute, CH-4002 Basel, Switzerland; c.dehoogh@swisstph.ch (K.d.H.); apolline.saucy@isglobal.org (A.S.); martin.roosli@swisstph.ch (M.R.); 3Faculty of Science, University of Basel, CH-4003 Basel, Switzerland; 4Barcelona Institute for Global Health, 08036 Barcelona, Spain; 5Department of Environmental Affairs and Developmental Planning, Western Cape Government, Cape Town 7925, South Africa; Bhawoodien.Parker@westerncape.gov.za (B.P.); joy.leaner@westerncape.gov.za (J.L.)

**Keywords:** ambient air pollution, cardiorespiratory outcomes, particulate matter, informal settlements, chest pain, adults

## Abstract

Few studies have investigated the relationship between ambient air pollution and cardiorespiratory outcomes in Africa. A cross-sectional study comprising of 572 adults from four informal settlements in the Western Cape, South Africa was conducted. Participants completed a questionnaire adapted from the European Community Respiratory Health Survey, and the National Health and Nutrition Examination Survey questionnaire. Exposure estimates were previously modelled using Land-Use Regression for Particulate Matter (PM_2.5_) and Nitrogen Dioxide (NO_2_) at participants’ homes. The median age of the participants was 40.7 years, and 88.5% were female. The median annual NO_2_ level was 19.7 µg/m^3^ (interquartile range [IQR: 9.6–23.7]) and the median annual PM_2.5_ level was 9.7 µg/m^3^ (IQR: 7.3–12.4). Logistic regression analysis was used to assess associations between outcome variables and air pollutants. An interquartile range increase of 5.12 µg/m^3^ in PM_2.5_ was significantly associated with an increased prevalence of self-reported chest-pain, [Odds ratio: 1.38 (95% CI: 1.06–1.80)], adjusting for NO_2_, and other covariates. The study found preliminary circumstantial evidence of an association between annual ambient PM_2.5_ exposure and self-reported chest-pain (a crude proxy of angina-related pain), even at levels below the South African National Ambient Air Quality Standards.

## 1. Introduction

In 2016, the World Health Organisation (WHO) estimated that 4.2 million deaths were attributed to ambient air pollution [1]. Furthermore, four air pollutants were found to have adverse effects on health, namely, particulate matter (PM), sulphur dioxide (SO_2_), nitrogen dioxide (NO_2_) and ozone (O_3_) [1]. Approximately 58% of the air pollution related deaths was due to ischaemic heart disease and stroke, 18% was due to chronic obstructive pulmonary disease and acute lower respiratory tract infections, respectively, and 6% was due to lung cancer [1]. In South Africa, air pollutants contributed to approximately 740,199 disability adjusted life years and 22971 deaths, all due to cardiopulmonary cancers and non-cancer diseases [2,3].

Despite the majority of ambient air pollution-related deaths occurring in low-and middle-income countries, most of the studies investigating the relationship between ambient air pollution and cardiorespiratory outcomes have been conducted in America, Asia and Europe where air pollution levels and composition as well as population characteristics differ from those in Africa and more specifically in South Africa’s informal settlements [1,4,5,6,7,8,9,10,11]. Furthermore, epidemiological studies conducted in Africa investigating this relationship have notable limitations, including a lack of robust exposure measurements, a lack of objective outcome measurements or inadequate adjustment of possible confounders [12,13,14,15,16].

There is therefore a paucity of robust data on the continent describing the relationship between ambient air pollution and cardiorespiratory outcomes, especially amongst adults residing in informal settlements, who might be disproportionately affected by air pollution due to underlying susceptibility and lifestyle behaviour in informal communities. This study aimed to determine the relationship between modelled annual exposure estimates of PM_2.5_ and NO_2_ concentrations with self-reported cardiorespiratory outcomes amongst adults residing in four informal settlements of the Western Cape province of South Africa.

This study involved the analysis of a subset of data that was collected as part of a larger cohort study conducted in 2016 investigating the association between ambient air quality and respiratory morbidities including childhood asthma among 590 primary school pupils in the Western Cape [16]. The current study is a cross-sectional study investigating the association between ambient air quality and self-reported cardiorespiratory outcomes among the parents/guardians of these primary school pupils.

## 2. Materials and Methods 

### 2.1. Study Design, Population and Sampling

A detailed description of the study area and methodology have been previously published [17]. In brief, the study areas included informal settlements in three areas identified in the Western Cape. These areas were selected to maximise contrasts in exposure levels to the different ambient air pollutants. The areas included an urban industrialised area (Marconi-Beam in Milnerton), a peri-urban area with a large informal sector (Khayelitsha) and a semi-arid rural area (Oudtshoorn). An additional area served as a background (with a low air pollution score ranking) and with comparable socio-economic status as the three identified areas (Masiphumelele in Noordhoek).

For this study, all the parents of the primary school pupils selected for the larger cohort study were included [17]. Briefly, the pupils were recruited from schools located near the ambient air quality monitoring stations in the selected areas. Approximately 150 pupils in each study area were selected. After meeting each school’s principal, obtaining permission from the school board and obtaining class lists and addresses, the houses of the primary school children were visited by trained field staff to obtain consent from the caregivers (parent or guardian). Consent was also obtained from the parents/guardians for their own participation in the sub-study. Only those who consented were included in the study. Only the parents/guardians of the pupils selected in the larger study were recruited for this current study. 

### 2.2. Questionnaire

Trained interviewers administered the questionnaires to participants in their spoken language (English, Xhosa or Afrikaans). The questionnaire was back-translated to ensure the consistency and reliability of the questionnaires. The use of mobile technology was implemented in the administration and capture of questionnaires.

The questionnaire used for this study included items on: demographic characteristics, residential history, respiratory health, cardiovascular disease, blood pressure and other chronic illnesses such as high cholesterol, occupational history, exposure to indoor pollutants, exposure to outdoor pollutants, physical activity, psychosocial stress and tobacco use.

Questions from the European Community Respiratory Health Survey [18], as well as the National Health and Nutrition Examination Survey questionnaire [19], were incorporated into this study’s questionnaire (Table 1). 

### 2.3. Outcome Characterisation

An asthma symptom score was created from the responses to eight (8) asthma-related questions (Table 1). One point was allocated for each positive response. A similar asthma symptom score, excluding the variable self-reported asthma, has been validated elsewhere [20]. Furthermore, additional cardiorespiratory self-reported outcomes include: doctor diagnosed asthma, chest-pain, hypertension, and high cholesterol (Table 1).

### 2.4. Exposure Characterisation

The annual average concentration of PM_2.5_ and NO_2_ was estimated at each participant’s address by land-use regression (LUR) models developed specifically for this study and have been previously published elsewhere. [21] In brief, the air pollution monitoring campaigns were performed during 2015–2016 in each study area. Weekly measurements of PM_2.5_ and NO_2_ were performed in both winter and summer at 140 sites (40 sites each in three study areas, and 20 sites in Masiphumulele) within a period of one year. These measurements were temporally adjusted using routinely monitored air quality measurements to obtain the seasonal (winter/summer) and annual averages. Predictors of exposure, obtained or collected on-site, such as household density, nearby traffic (e.g., major roads, bus stops, and train stations), waste burning sites, and land-use derived from geographic information system (GIS) were used to evaluate the spatial variation in the annual average concentrations. To maximize the adjusted explained variance, regression models were developed, using a supervised stepwise approach, and the models were validated using leave-one-out-cross-validation (LOOCV). The LUR models were used to estimate annual average concentration of PM_2.5_ and NO_2_ at each participant’s address. The annual NO_2_ LUR model explained 76% of the spatial variability in the NO_2_ adjusted concentrations, 62% for the warm dry summer season and 77% for the cold and wet winter season. The annual PM_2.5_ LUR model explained 29% of the spatial variability in the PM_2.5_ adjusted concentrations, 36% for the warm season and 29% for the cold season.

### 2.5. Statistical Analysis

Data was captured, cleaned and analysed in Stata: Release 11 (StataCorp. 2009. Statistical Software. StataCorp LP: College Station, TX, USA)**.** Descriptive statistics were used to examine the characteristics of the study population, cardiorespiratory outcomes and various potential confounders. Bivariate regression was used to measure associations between various potential confounders and cardiorespiratory outcomes. Relevant confounders were identified a priori or based on bivariate associations, and they were incrementally included in multivariable logistic regression models (age, sex, use of paraffin, smoking, physical activity, and study area). The base model was a single-pollutant model comprising cardiorespiratory outcomes as the dependent variable with either annual NO_2_ or annual PM_2.5_ as the independent variable. The final model was a two-pollutant model, which included cardiorespiratory outcomes as the dependant variable and both, annual NO_2_ and annual PM_2.5_ as independent variables.

### 2.6. Ethical Considerations

The main study was approved by the University of Cape Town’s Research Ethics Committee (ethics number: 234/2009). The protocol for the sub-study was approved by the University of Cape Town’s Research Ethics Committee (ethics number: 639/2018).

## 3. Results

### 3.1. Demographic Characteristics of the Study Participants

Table 2 shows that the majority of participants were female (88.5%), that 75% of the participants completed high school, and that the majority of the participants spoke isiXhosa (69.6%). Masiphumelele (Noordhoek) had the highest proportion of employed participants (51.8%). The participants from Masiphumelele were also younger (median age, 38.2 years) than those from the other study areas. More than half (61.4%) of the participants made use of paraffin for cooking and heating. Khayelitsha had the highest proportion of participants who were physically active in the last month before the interview (31.4%), while Oudtshoorn had the highest proportion of participants who smoked cigarettes (32.1%).

### 3.2. Cardiorespiratory Outcomes of the Study Participants

The prevalence of self-reported doctor-diagnosed asthma was 6.6%. Khayelitsha had the highest proportion of participants who reported having experienced wheezing (13.4%), shortness of breath (10.5%) and tight chest (12.2%) in the last 12 months (Table 3). Participants from Masiphumelele (Noordhoek) had the highest proportion of participants who reported having experienced shortness of breath after exercise (25.9%) and Oudtshoorn had the highest proportion of participants bringing up phlegm from the chest during winter (12.2%). Furthermore, all the study areas had participants who experienced at least one asthma symptom in the preceding year. Participants from Khayelitsha had the highest prevalence of self-reported chest-pain (14.5%) and self-reported cholesterol (8.7%). Khayelitsha and Oudtshoorn had the highest proportion of participants with self-reported hypertension (24.4%).

### 3.3. Air Pollution Characterization Based on LUR Modelling

Across the four study areas, the estimated mean annual NO_2_ concentration was 16.9 µg/m^3^ (interquartile range: 9.6 µg/m^3^ to 23.7 µg/m^3^) and the estimated mean annual PM_2.5_ concentration was 10.1 µg/m^3^ (interquartile range: 7.3 µg/m^3^ to 12.4 µg/m^3^). Furthermore, the estimated mean annual concentrations suggest that participants residing in Khayelitsha were exposed to the highest levels of both NO_2_ and PM_2.5_ (Figure 1). Masiphumulele had the lowest variability in estimated annual average NO_2_ and PM_2.5_ concentrations (Figure 1). In addition, the annual estimated annual average NO_2_ and PM_2.5_ concentrations were slightly positively correlated (rho = 0.32).

### 3.4. Association between NO_2_ and PM_2.5_ Levels and Self-Reported Reported Cardiorespiratory Outcomes

An interquartile range increase of 5.12 µg/m^3^ in PM_2.5_ was statistically significantly associated with an increase in prevalence of self-reported chest-pain, both in the single—[odds ratio: 1.36 (95% CI: 1.05–1.78)] (Table 4 model F) and two-pollutant model [odds ratio: 1.38 (95% CI: 1.06–1.80)] (Table 5 model F), which included adjustment for NO_2_, and other covariates. There was also a marginal statistically significant positive association of PM_2.5_ with both doctor-diagnosed asthma and more than two asthma symptom score (ASS > 2), in both the single- and two-pollutant models (Table 4 and Table 5 model F). However, there was no significant association observed for NO_2_ with any of the outcomes assessed. A sensitivity analysis excluding males did not change the associations found (results not shown). 

## 4. Discussion

This cross-sectional study conducted amongst adults from four informal settlements in the Western Cape province of South Africa found an interquartile range increase of 5.12 µg/m^3^ in PM_2.5_ to be statistically significantly associated with an increase prevalence of self-reported chest-pain, both in the single—[1.36 (95% CI: 1.05–1.78)] and two-pollutant model [1.38 (95% CI: 1.06–1.80)], which controlled for NO_2_. However, none of the outcomes investigated were significantly associated with NO_2_.

All positive associations (both those statistically significant and those not significant) with PM_2.5_ in this study were at concentrations lower than National Ambient Air Quality Standards (NAAQS). For instance, the average annual estimated PM_2.5_ concentration of 10.1 µg/m^3^ was lower than the South African NAAQS of 25 µg/m^3^, while it was higher than the World Health Organisation (WHO) Air Quality Guideline of 5 µg/m^3^ [1,22,23]. The average annual estimated NO_2_ concentration of 16.9 µg/m^3^ was also lower than the South African NAAQS [22].

In comparison to previous studies within Africa, the estimated annual PM_2.5_ concentrations in this study were similar to those measured using stationary monitors in Zambia (10.2 µg/m^3^), while it was lower than measurements obtained in Zimbabwe (40.6 µg/m^3^), and in a previous study conducted in Bethlehem in South Africa (65 µg/m^3^) [24,25,26]. Moreover, the estimated annual NO_2_ levels in the current study were lower than levels measured in a previous study in Cape Town (25.1 µg/m^3^) [13]. 

Only two previous studies conducted in Africa that measured the prevalence of at least one self-reported cardiorespiratory outcome as reported in this current study, could be found. The prevalence of self-reported asthma among residents in Windhoek (11%) [16] in which the participants were all younger than 30 years old, was higher compared to this current study (6.6%), while self-reported hypertension (5.6%) among the elderly (55 years old and older) living near mine dumps in Gauteng and North West, was considerably lower than in this current study (20.1%) [27]. 

Associations between air pollutants and adverse cardiovascular outcomes have been found in a number of international studies [4,5,6,7,8,28,29,30,31,32,33] and a previous study in Cape Town [13]. These studies differed from the current study in one or more respects such the health outcome (mortality or hospital admissions), PM exposure being short-term and >25.1 µg/m^3^ and air pollution measured from stationary monitors. 

A study conducted in Canada involving more than 5 million participants found a 3.5 µg/m^3^ interquartile range increase in PM_2.5_ to be associated with 1.04 (95% CI: 1.03–1.05) risk of acute myocardial infarction [33]. The estimated mean PM_2.5_ concentration of 9.6 µg/m^3^ in the latter study was similar to the annual estimated PM_2.5_ levels in the current study. It is thus likely that adverse cardiovascular effects can occur at PM_2.5_ levels that comply with the thresholds of the NAAQS. However, it should also be noted that there are non-cardiac causes of chest pain and so the likely association observed in the current study could likely reflect adverse effects from other systems [34]. In the study conducted in Gauteng and North West among elderly poor communities, those living near mine dumps (1–2 km) had a significantly higher prevalence of cardiovascular disease than those living further away (≥5 km). A cohort of 21 countries, found an association between increased PM_2.5_ levels and cardiovascular disease mortality and the effect in low-and middle-income countries was similar to that of communities with PM_2.5_ levels > 35 μg/m^3^ [35].

This study, to the best of our knowledge, is the only to have been conducted on the African continent that used Land-Use Regression to estimate annual exposure to ambient air pollution to assess its relationship with cardiorespiratory outcomes among adults residing in informal settlements. Land-use regression models were used to estimate each participant’s annual concentration of exposure to NO_2_ and PM_2.5_ at their current address during the study period. We assumed that these annual exposures were not temporally variable in previous years prior to the study. The annual NO_2_ LUR model explained 76% of the spatial variability in the NO_2_ adjusted concentrations, 62% for the warm dry summer season and 77% for the cold and wet winter season. The annual PM_2.5_ LUR model explained 29% of the spatial variability in the PM_2.5_ adjusted concentrations, 36% for the warm season and 29% for the cold season [21]. However, the lower predictive power of the PM_2.5_ model could have been caused by non-differential exposure misclassification, thereby biasing the association with the other outcomes investigated towards the null (reduced the significance of the association). 

However, there are several limitations in the current study. Outcomes were self-reported symptoms, and objective measurements such as lung function testing and forced-exhaled nitric oxide were not available in the study. Such self-reporting could have introduced both recall and reporting biases and may have introduced significant outcome misclassification. It is possible that this non-differential outcome misclassification may thus bias the association between air pollutants and respiratory outcomes towards the null (reduced the significance of the association). Additionally, the less-than-ideal exposure estimation in the study with low exposure variability for PM _2.5_ (IQR ≈ 5 µg/m^3^) and a model with R^2^ ≈ 0.29 to generate exposure predictions could lead to bias to the null [36]. Although data collected from several local sources such as housing and population density, waste burning, outdoor grilling, bus routes, and construction sites were found to be predictors of PM _2.5_ concentrations, a lack of comprehensive data on such sources, especially their transient nature, could explain a significant part of the low variability found for PM_2.5_ [21]. The low PM_2.5_ variability in the study is not necessarily unexpected considering the small scale of the four study neighbourhoods (<40 km^2^) and the relatively small distances between the four sites (<500 km). PM_2.5_ mostly reflects background pollution with low small-scale variability. The cross-sectional design of the study limits inference of the association to prevalence, and the temporality of such association cannot be ascertained. In addition, with more than four-fifths of the study participants being women, it is difficult to generalize the findings to men nor the general population. Lastly, the study was not sufficiently powered to detect most associations. 

## 5. Conclusions

This study provided some preliminary evidence of the association between ambient PM_2.5_ levels and the prevalence of self-reported cardiovascular morbidity (chest-pain) at levels below the NAAQS. The presence of an association below the NAAQS supports the need to revisit it to determine its efficacy in terms of protecting vulnerable population, especially those from informal settlements who may have underlying vulnerability and/or are disproportionately affected by the burden of air pollution. It is important to conduct further research amongst this population using a more objective outcome with a longitudinal study design.

## Figures and Tables

**Figure 1 ijerph-18-13306-f001:**
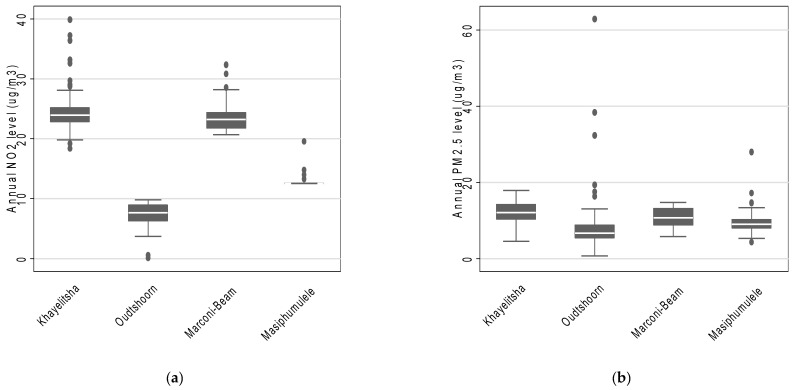
The distribution of annual estimates of (**a**) NO_2_ and (**b**) PM_2.5_ concentrations across the four study areas based on LUR modelling.

**Table 1 ijerph-18-13306-t001:** Cardiorespiratory questions included in the questionnaire.

**Asthma-Related Questions**
Wheezing in the last 12 months
Shortness of breath in the last 12 months
Woken up by feeling of tight chest in the last 12 months
Attack of shortness of breath at rest in the last 12 months
Woken up by attack of shortness of breath in the last 12 months
Attack of shortness of breath after exercise in the last 12 months
Self- reported asthma
Medication for asthma control
**Additional Cardiorespiratory Questions**
Self-reported doctor diagnosis of asthma
Self-reported chest-pain
Self-reported hypertension
Self-reported high cholesterol

**Table 2 ijerph-18-13306-t002:** Demographics and selected characteristics of the study participants across the four study areas.

	Khayelitsha (N = 172)	Marconi-Beam (N = 132)	Oudtshoorn (N = 156)	Masiphumelele (N = 112)	All (N = 572)
Age	40.8 (35.4; 47.5)	38.8 (32.7; 43.6)	43.0 (36.3; 51.5)	38.2 (33.8; 45.2)	40.7 (34.6; 47.2)
Sex (female)	151 (87.8)	121 (91.7)	142 (91.0)	92 (82.1)	506 (88.5)
Home language isiXhosa English, Afrikaans or other	148 (86.0) 38 (22.1)	110 (83.3) 33 (25.0)	32 (20.5) 124 (79.5)	108 (96.4) 9 (8.0)	398 (69.6) 204 (35.7)
Education Never attended school Attended preschool, primary school or high school Completed high school	2 (1.2) 169 (98.3) 124 (72.1)	2 (1.5) 130 (98.5) 94 (71.2)	0 (0) 156 (100) 119 (76.5)	0 (0) 112 (100) 93 (83.0)	4 (0.7) 567 (99.1) 429 (75.0)
Employed	77 (44.8)	57 (43.2)	50 (32.1)	58 (51.8)	242 (42.3)
Fuel used to heat home or cooking Wood Coal Gas Paraffin Animal dung Solar energy Electricity	3 (1.7) 0 (0) 16 (9.3) 140 (81.4) 0 (0) 0 (0) 82 (47.7)	0 (0) 0 (0) 4 (3.0) 108 (81.8) 0 (0) 0 (0) 52 (39.4)	6 (3.9) 0 (0) 9 (5.8) 33 (21.2) 0 (0) 0 (0) 131 (84.0)	0 (0) 1 (0.9) 3 (2.7) 70 (62.5) 0 (0) 0 (0) 54 (48.2)	9 (1.6) 1 (0.2) 32 (5.6) 351 (61.4) 0 (0) 0 (0) 319 (55.8)
Physical active in last month	54 (31.4)	36 (27.3)	31 (19.9)	22 (19.6)	143 (25.0)
Smoke cigarettes	16 (9.3)	3 (2.3)	50 (32.1)	7 (6.3)	76 (13.3)

Categorical variables depicted as number (%), numerical variables depicted as median (25th percentile, 75th percentile).

**Table 3 ijerph-18-13306-t003:** Cardiorespiratory outcomes of the study participants across the four study areas.

	Khayelitsha (N = 172) N (%)	Marconi-Beam (N = 132) N (%)	Oudtshoorn (N = 156) N (%)	Masiphumelele (N = 112) N (%)	All areas (N = 572) N (%)
Doctor diagnosed asthma	14 (8.1)	7 (5.3)	11 (7.1)	6 (5.4)	38 (6.6)
Wheezing in the last 12 months	23 (13.4)	8 (6.1)	13 (8.3)	11 (9.8)	55 (9.6)
Shortness of breath in last 12 months	18 (10.5)	7 (5.3)	14 (9.0)	7 (6.3)	46 (8.1)
Woken up by feeling of tight chest in the last 12 months	21 (12.2)	9 (6.8)	14 (9.0)	9 (8.0)	53 (9.3)
Woken up by attack of shortness of breath at rest in the last 12 months	18 (10.5)	7 (5.3)	14 (9.0)	9 (8.0)	48 (8.4)
Attack of shortness of breath after exercise in the last 12 months	35 (20.3)	14 (10.6)	26 (16.7)	29 (25.9)	104 (18.2)
Bring up phlegm from chest at any time of day in the winter	20 (11.6)	3 (2.3)	19 (12.2)	10 (8.9)	52 (9.1)
Woken up by heavy coughing at any time in the last 12 months	29 (16.9)	6 (4.5)	22 (14.1)	16 (14.3)	73 (12.8)
Self-reported asthmatic	16 (9.3)	11 (8.3)	11 (7.1)	6 (5.4)	44 (7.7)
Medication for asthma control	7 (4.1)	2 (1.5)	10 (6.4)	1 (0.9)	20 (3.5)
Asthma symptom score ^#^ Score = 0 Score = 1 Score > 1	0 (0) 108 (62.8) 78 (45.3)	0 (0) 108 (81.8) 35 (26.5)	0 (0) 123 (78.8) 33 (21.2)	0 (0) 72 (64.3) 45 (40.2)	0 (0) 411 (71.9) 191(33.4)
Self-reported experience chest pain	25 (14.5)	5 (3.8)	17 (10.9)	4 (3.6)	51 (8.9)
Self-reported hypertension	42 (24.4)	14 (10.6)	38 (24.4)	21 (18.8)	115 (20.1)
Self-reported cholesterol	15 (8.7)	8 (6.1)	5 (3.2)	5 (4.5)	33 (5.8)

# Asthma symptom score calculated as sum of: Wheezing in the last 12 months, shortness of breath in last 12 months; woken up by feeling of tight chest in the last 12 months; attack of shortness of breath at rest in the last 12 months; attack of shortness of breath after exercise in the last 12 months; woken up by attack of shortness of breath in the last 12 months; self-reported asthma; medication for asthma control.

**Table 4 ijerph-18-13306-t004:** Association between interquartile increase in estimated annual NO_2_ and PM_2.5_ levels and self-reported cardiorespiratory outcomes in single-pollutant model (reported as odds ratios (95% confidence interval).

	NO_2_					PM_2.5_				
	Doctor Diagnosed Asthma	Asthma Symptom Score > 2	Chest-Pain	Hypertension	High Cholesterol	Doctor Diagnosed Asthma	Asthma Symptom Score > 2	Chest-Pain	Hypertension	High Cholesterol
A	0.92 (0.48–1.75)	1.31 (0.92–1.85)	1.06 (0.61–1.83)	0.75 (0.50–1.11)	1.90 (0.94–3.83)	1.21 (0.90–1.62)	1.14 (0.93–1.41)	**1.38 (1.06–1.80)**	0.90 (0.69–1.18)	1.15 (0.83–1.60)
B	1.13 (0.58–2.19)	1.40 (0.98–1.99)	1.15 (0.66–2.02)	0.93 (0.61–1.42)	**2.29 (1.11–4.72)**	1.26 (0.95–1.69)	1.16 (0.94–1.43)	**1.41 (1.08–1.85)**	0.95 (0.73–1.25)	1.20 (0.86–1.66)
C	1.61 (0.76–3.41)	1.23 (0.81–1.87)	1.23 (0.65–2.35)	0.83 (0.50–1.39)	1.98 (0.84–4.64)	1.28 (0.97–1.69)	1.14 (0.92–1.41)	**1.42 (1.08–1.86)**	0.94 (0.70–1.25)	1.16 (0.79–1.69)
D	1.67 (0.78–3.55)	1.34 (0.88–2.05)	1.42 (0.74–2.74)	0.87 (0.52–1.45)	2.17 (0.93–5.11)	1.28 (0.97–1.69)	1.15 (0.93–1.42)	**1.42 (1.09–1.86)**	0.95 (0.71–1.26)	1.18 (0.82–1.72)
E	1.66 (0.78–3.55)	1.43 (0.93–2.20)	1.45 (0.75–2.79)	0.89 (0.53–1.49)	2.24 (0.96–5.22)	1.28 (0.97–1.69)	1.17 (0.95–1.46)	**1.43 (1.09–1.87)**	0.96 (0.73–1.27)	1.20 (0.84–1.71)
F	1.13 (0.11–12.05)	1.23 (0.34–4.46)	0.65 (0.10–4.33)	0.78 (0.18–3.49)	0.40 (0.03–5.83)	1.27 (0.95–1.71)	1.12 (0.89–1.41)	**1.36 (1.05–1.78)**	0.96 (0.72–1.29)	1.13 (0.68–1.86)

A—Base model (NO_2_ or PM_2.5_); B—Base model + Age + Sex.; C—Base model + Age + Sex + Paraffin.; D—Base model + Age + Sex + Paraffin + Smoking.; E—Base model + Age + Sex + Paraffin + Smoking + Physical activity.; F—Base model + Age + Sex + Paraffin + Smoking + Physical activity + Area.; bold denotes significance at the 0.05 level of significance.

**Table 5 ijerph-18-13306-t005:** Association between interquartile increase in estimated annual NO_2_ and PM_2.5_ levels and self-reported cardiorespiratory outcomes in two-pollutant model, reported as odds ratios (95% confidence intervals).

	NO_2_					PM_2.5_				
	Doctor Diagnosed Asthma	Asthma Symptom Score > 2	Chest-Pain	Hypertension	High Cholesterol	Doctor Diagnosed Asthma	Asthma Symptom Score > 2	Chest-Pain	Hypertension	High Cholesterol
A	0.84 (0.44–1.62)	1.24 (0.86–1.78)	0.93 (0.53–1.63)	0.76 (0.50–1.15)	1.83 (0.88–3.79)	1.23 (0.92–1.64)	1.11 (0.90–1.39)	**1.40 (1.06–1.84)**	0.97 (0.75–1.27)	1.08 (0.71–1.66)
B	1.03 (0.53–2.00)	1.32 (0.92–1.92)	1.00 (0.57–1.79)	0.94 (0.60–1.47)	**2.19 (1.03–4.64)**	1.27 (0.95–1.70)	1.12 (0.90–1.39)	**1.42 (1.08–1.88)**	0.98 (0.74–1.30)	1.10 (0.70–1.74)
C	1.47 (0.69–3.14)	1.16 (0.75–1.79)	1.07 (0.55–2.08)	0.84 (0.50–1.43)	1.90 (0.79–4.57)	1.27 (0.95–1.69)	1.13 (0.91–1.41)	**1.42 (1.08–1.88)**	0.98 (0.74–1.30)	1.09 (0.68–1.74)
D	1.53 (0.71–3.29)	1.26 (0.82–1.96)	1.26 (0.64–2.46)	0.88 (0.52–1.49)	2.07 (0.87–4.97)	1.27 (0.95–1.69)	1.13 (0.91–1.41)	**1.42 (1.09–1.86)**	0.98 (0.74–1.30)	1.12 (0.71–1.77)
E	1.53 (0.71–3.30)	1.34 (0.86–2.09)	1.29 (0.66–2.52)	0.89 (0.53–1.52)	2.13 (0.90–5.03)	1.27 (0.95–1.69)	1.15 (0.92–1.44)	**1.42 (1.09–1.87)**	0.99 (0.75–1.31)	1.15 (0.76–1.75)
F	1.10 (0.10–12.49)	1.21 (0.33–4.41)	0.58 (0.08–4.21)	0.78 (0.18–3.48)	0.39 (0.03–5.91)	1.28 (0.95–1.72)	1.13 (0.90–1.42)	**1.38 (1.06–1.80)**	0.97 (0.73–1.30)	1.14 (0.69–1.87)

A—Base model (NO_2_ and PM_2.5_); B—Base model + Age + Sex; C—Base model + Age + Sex + Paraffin; D—Base model + Age + Sex + Paraffin + Smoking; E—Base model + Age + Sex + Paraffin + Smoking + Physical activity; F—Base model + Age + Sex + Paraffin + Smoking + Physical activity + Area; bold denotes significance at the 0.05 level of significance.

## Data Availability

The data that support the findings of this study are available from the corresponding author upon reasonable request and with permission of DEA&DP, South Africa in a form which ensures privacy of study participants.

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
