# Peer review of "Ambient Air Pollution and Cardiorespiratory Outcomes amongst Adults Residing in Four Informal Settlements in the Western Province of South Africa"

_ijerph, 2021, doi:10.3390/ijerph182413306_

Round 1

Reviewer 1 Report

The approach presented in the manuscript is novel since most investigations on ambient air pollution are carried out at well-established urban communities. This piece of research will contribute significant knowledge understanding the impacts of ambient air pollution at informal settlements.

I am just attaching a word file with minor suggestions concerning graphs and tables.

Author Response

  1. I suggest a homogeneous scale (0-60) for the Y axis in order to facilitate comparison, as well as specification of measurement units. It is also recommended to differentiate each graph as a and b. For example a (NO2) and b (PM2.5)

Response: The Y axis scale of Figure 1 was changed to 0-60 and graphs labelled as “A” and “B)” as suggested by the reviewer to allow for easier comparison between the two graphs (section 3.3)

  1. I suggest rearranging results in the same row. Example A 0.92 (0.48 -1.75), etc. I feel your data will communicate better

Response: The font in Table 4 was reduced to fit the results in the same row as suggested by the reviewer. (Section 3.4)

  1. I suggest the same as above note

Response: The font in Table 5 was reduced to fit the results in the same row as suggested by the reviewer. (Section 3.4)

Reviewer 2 Report

  • The abstract is misleading, where it seems that the data used is developed in this manuscript, where the data was modeled and published previously. 
  • The introduction is short, needs more relevant information, 
  • Material and Methods: Please move the first seven lines to the introduction from “This study involved the analysis” to “the parents/guardians of these primary school pupils.”. These sentences belong in the introduction and outline of the objective of the study which should be in the last paragraph.
  • Figure 1: Please add units to the y-axis. 

Author Response

  1. The abstract is misleading, where it seems that the data used is developed in this manuscript, where the data was modeled and published previously

Response: The fourth sentence in the abstract was revised to read “Exposure estimates were previously modelled using Land-Use Regression for Particulate Matter (PM2.5) and Nitrogen Dioxide (NO2) at participant’s home.”

  1. The introduction is short, needs more relevant information

Response: The introduction was revised to include more relevant information. Additionally, the first seven lines in Materials and Methods was also moved to the Introduction (section 1 last paragraph). The revised text reads:

“In 2016, the World Health Organisation (WHO) estimated that 4.2 million deaths were attributed to ambient air pollution1. Furthermore, four air pollutants were found to have adverse effects on health, namely, Particulate Matter (PM), Sulphur Dioxide (SO2), Nitrogen Dioxide (NO2) and Ozone (O3).1 Approximately 58% of the air pollution related deaths was due to ischaemic heart disease and stroke, 18% was due to chronic obstructive pulmonary disease and acute lower respiratory tract infections respectively and 6% was due to lung cancer1. In South Africa, air pollutants contributed to approximately 740199 disability adjusted life years and 22 971 deaths, all due to cardiopulmonary cancers and non-cancer diseases.2, 3 

Despite the majority of ambient air pollution-related deaths occurring in low-and middle-income countries, most of the studies investigating the relationship between ambient air pollution and cardiorespiratory outcomes have been conducted in America, Asia and Europe where air pollution levels and composition as well as population characteristics differ from those in Africa and more specifically in South Africa’s informal settlements.1,4-11  Furthermore, epidemiological studies conducted in Africa investigating this relationship have notable limitations including a lack of robust exposure measurements, a lack of objective outcome measurements or inadequate adjustment of possible confounders.12-16

There is therefore a paucity of robust data on the continent describing the relationship between ambient air pollution and cardiorespiratory outcomes, especially amongst adults residing in informal settlements, who might be disproportionately affected by air pollution due to underlying susceptibility and lifestyle behavior in informal communities. This study aimed to determine the relationship between modelled annual exposure estimates of PM2.5 and NO2 concentrations with self-reported cardiorespiratory outcomes amongst adults residing in four informal settlements of the Western Cape province of South Africa.This study involved the analysis of a subset of data that was collected as part of a larger cohort study conducted in 2016 investigating the association between ambient air quality and respiratory morbidities including childhood asthma among 590 primary school pupils in the Western Cape.16 The current study is a cross-sectional study investigating the association between ambient air quality and self-reported cardiorespiratory outcomes among the parents/guardians of these primary school pupils.”

  1. Material and Methods: Please move the first seven lines to the introduction from “This study involved the analysis” to “the parents/guardians of these primary school pupils.”. These sentences belong in the introduction and outline of the objective of the study which should be in the last paragraph

Response: The revision suggested was made as indicated in the response to the previous comment.

  1. Figure 1: Please add units to the y-axis.

Response: µg/m3 added to the y-axis  (section 3.3 figure 1)

Reviewer 3 Report

This paper reports on PM2.5 and NO2 exposure and cross-sectional associations with cardiorespiratory outcomes among adults living in informal settlements in the Western Province of South Africa. Pollutant exposures were modeled with land use regression to estimate annual average concentration at each participant’s residence. Logistic regression models were used to characterize the relationship between pollutant exposures and health outcomes. The study adds to the limited research on this area and seems generally well-conducted and described. I have a few suggestions and questions for the authors listed below.

Abstract:

-I recommend authors state in the abstract that the pollutant measures are annual averages when providing those concentrations.

-I suggest including the type of statistical model in the abstract (logistic regression), and instead of reporting “regression coefficients”, authors should revise by transforming (e.g., exponentiating, if applicable) and interpreting regression coefficients as odd ratios, e.g., “an IQR increase of 5.12 µg/m3 PM2.5 was associated with an X increase in the odds of…”  

-I suggest adding the cross-sectional study design, as mentioned in the Methods section, to the abstract also.

Methods:

-In Section 2.4, Exposure Characterization, I recommend reiterating the major performance measures from Saucy et al., 2018 (e.g. R2 = 0.76 and 0.29 for annual NO2 and PM2.5 model, respectively).

-Section 2.5, Statistical Analysis. I don’t really understand the last sentence and why in a two pollutant model if both NO2 and PM2.5 are included the distinction “…either annual NO2 (and PM2.5 as a co-exposure) or annual PM2.5 (and NO2 as a co-exposure) as independent variables” is being made. If both pollutants are included, then could this be more simply stated “or a two-pollutant model which included cardiorespiratory outcomes as the dependent variable and NO2 and PM2.5 as the independent variables”?

Results:

-The statement “99.1% of the participants had completed preschool, primary school or high school education” does not provide enough information to be meaningful. If the point is 99.1 percent of participants had any education, I would say that, but right now, readers have no way of knowing how many participants completed high school. Also, Table 2 says “attended,” which is not the same as “completed”. On a related note, especially if this is the finest resolution of education available, is there a variable related to individual or household income?

-Given 88.5% of participants are female, did the authors consider a sensitivity analysis restricted to females?

-Section 3.4 Authors should be clear about the results they are discussing in the text comes from, for example the fully adjusted model D in Tables 4 and 5 rather than just “other covariates.”

Discussion:

-paragraph 3 discussion standards. Authors should provide a reference for these standards, and explicitly state it is the South African standard since other countries use a similar acronym but have different values (e.g. US EPA NAAQS). Also, didn’t the WHO just update its PM2.5 guidance to an annual average of 5 µg/m3? I’m also confused by reference 21 (Suid-afrika RVAN. National Ambient Air Quality Standards [Internet]. Government Gazette; 2009. p. 4. Available from: 160 http://www.epa.gov/air/criteria.html) I expect a link to a South African government website, not the US EPA – is that a mistake?

-I appreciate the authors’ attempt to draw comparisons to the available literature (e.g., paragraph 4 of the discussion) are made difficult by a lack of comparison studies. However, some more details from the studies mentioned might be helpful, for example, the average age of the participants in the referenced study. In addition, authors could draw on less-than-ideal studies for comparison. For example, there might not be any studies on these outcomes in traditional housing/settlements in South Africa, but what about other countries? It would help provide some context for the results and if, as the authors state, those living in informal settlements may be uniquely susceptible.

-There are some several points of concern related to the pollutant estimates used by the authors in the epidemiologic analysis. First, there is not a great deal of exposure variability for PM2.5 (IQR ~ 5 µg/m3) and authors should acknowledge this. Second and perhaps more importantly, is the low LUR model performance used to generate the PM2.5 exposure estimates. The authors acknowledge this in the discussion, however, it still seems hard for me to reconcile using a model with R2 ≈ 0.29 to generate exposure predictions for an environmental epi study. More attention should be devoted to these issues in the discussion of the observed results. Also, authors may wish to place their assumptions that typical/traditionally worse exposure measurement lead to bias to the null, in context with studies such as Szpiro et al., 2011; Sheppard et al., 2012; van Smeden et al., 2020; and others.

Szpiro AA, Paciorek CJ, Sheppard L. Does more accurate exposure prediction necessarily improve health effect estimates?. Epidemiology (Cambridge, Mass.). 2011 Sep;22(5):680.

Sheppard L, Burnett RT, Szpiro AA, Kim SY, Jerrett M, Pope CA, Brunekreef B. Confounding and exposure measurement error in air pollution epidemiology. Air Quality, Atmosphere & Health. 2012 Jun;5(2):203-16.

van Smeden M, Lash TL, Groenwold RH. Reflection on modern methods: five myths about measurement error in epidemiological research. International journal of epidemiology. 2020 Feb 1;49(1):338-47.

Author Response

  1. I recommend authors state in the abstract that the pollutant measures are annual averages when providing those concentrations

Response: The abstract was revised to indicate that pollutant measures are annual averages (sentences 3 and 4).

  1. I suggest including the type of statistical model in the abstract (logistic regression), and instead of reporting “regression coefficients”, authors should revise by transforming (e.g., exponentiating, if applicable) and interpreting regression coefficients as odd ratios, e.g., “an IQR increase of 5.12 µg/m3 PM2.5 was associated with an X increase in the odds of…”  

Response: The abstract was revised by indicating that logistic regression was used and including odds ratios as suggested by the reviewer. (Third and second last sentence in the abstract)

  1. I suggest adding the cross-sectional study design, as mentioned in the Methods section, to the abstract also

Response: The Abstract and Introduction sections was revised to state that a cross sectional study design was used (Second sentence in abstract and last sentence in the introduction)

  1. In Section 2.4, Exposure Characterization, I recommend reiterating the major performance measures from Saucy et al., 2018 (e.g. R2 = 0.76 and 0.29 for annual NO2 and PM2.5 model, respectively

     Response: The last 2 sentences of section 2.4 was revised to read “The annual NO2 LUR       

     model explained 76% of the spatial variability in the NO2 adjusted concentrations, 62%

     for the warm dry summer season and 77% for the cold and wet winter season.  The

     annual PM2.5 LUR model explained 29% of the spatial variability in the PM2.5 adjusted

     concentrations, 36% for the warm season and 29% for the cold season”.

  1. Section 2.5, Statistical Analysis. I don’t really understand the last sentence and why in a two pollutant model if both NO2 and PM2.5 are included the distinction “…either annual NO2 (and PM2.5 as a co-exposure) or annual PM2.5 (and NO2 as a co-exposure) as independent variables” is being made. If both pollutants are included, then could this be more simply stated “or a two-pollutant model which included cardiorespiratory outcomes as the dependent variable and NO2 and PM2.5 as the independent variables”?

Response: The last sentence in the Statistical Analysis section was revised to state “

  1. The statement “99.1% of the participants had completed preschool, primary school or high school education” does not provide enough information to be meaningful. If the point is 99.1 percent of participants had any education, I would say that, but right now, readers have no way of knowing how many participants completed high school. Also, Table 2 says “attended,” which is not the same as “completed”. On a related note, especially if this is the finest resolution of education available, is there a variable related to individual or household income

Response: The number of participants that completed high school was included in Table 2. The sentence in section 3.1 was revised to read “Table 2 shows that the majority of participants were female (88.5%), that 75% of the participants completed high school and that the majority of the participants spoke isiXhosa (69.6%).

The question on household income was not well answered.

  1. Given 88.5% of participants are female, did the authors consider a sensitivity analysis restricted to females?

Response: A sensitivity analysis was performed excluding the males but this did not change the association found. The following sentence was added to Section 3.4 (last sentence)” A sensitivity analysis excluding males did not change the associations found (results not shown).”  

  1. Section 3.4 Authors should be clear about the results they are discussing in the text comes from, for example the fully adjusted model D in Tables 4 and 5 rather than just “other covariates.”

Response: The text revised to indicate that models referred to is model F in both tables (section Sentence 1 and 2.)

  1. paragraph 3 discussion standards. Authors should provide a reference for these standards, and explicitly state it is the South African standard since other countries use a similar acronym but have different values (e.g. US EPA NAAQS). Also, didn’t the WHO just update its PM2.5 guidance to an annual average of 5 µg/m3? I’m also confused by reference 21 (Suid-afrika RVAN. National Ambient Air Quality Standards [Internet]. Government Gazette; 2009. p. 4. Available from: 160 http://www.epa.gov/air/criteria.html) I expect a link to a South African government website, not the US EPA – is that a mistake?

Response: The text in paragraph 2 of the Discussion section was revised as suggested by the reviewer, the references cited and the link to the South African Air Quality Standards, corrected (https://www.gov.za/sites/default/files/gcis_document/201409/35463gon486.pdf). The text in the paragraph now reads”

     All positive associations (both those statistically significant and those not significant) with       

     PM2.5 in this study were at concentrations lower than National Ambient Air Quality

     Standards (NAAQS). For instance, the average annual estimated PM2.5 concentration of

     10.1µg/m3 was lower than the South African NAAQS of 25µg/m3, while it was higher than

     the World Health Organisation (WHO) Air Quality Guideline of 5µg/m3. 21, 22 The average

     annual estimated NO2 concentration of 16.9 µg/m3 was also lower than the South African

     NAAQS. 21

  1. I appreciate the authors’ attempt to draw comparisons to the available literature (e.g., paragraph 4 of the discussion) are made difficult by a lack of comparison studies. However, some more details from the studies mentioned might be helpful, for example, the average age of the participants in the referenced study. In addition, authors could draw on less-than-ideal studies for comparison. For example, there might not be any studies on these outcomes in traditional housing/settlements in South Africa, but what about other countries? It would help provide some context for the results and if, as the authors state, those living in informal settlements may be uniquely susceptible

Response: The ages of study participants were included in the comparative studies in the second sentence of the 4th paragraph of the Discussion. Details of reference studies were also included. Furthermore, comparisons  to studies in poor communities were included. The text read “In the study conducted in Gauteng and North West among elderly poor communities, those living near mine dumps (1-2 km) had a significantly higher prevalence of cardiovascular disease than those living further away (≥ 5 km). A cohort of 21 countries, found an association between increased PM2.5 levels and cardiovascular disease mortality and the effect in low-and middle-income countries were similar to that of communities with PM2.5 levels >35 μg/m3..35“(section 4 paragraph 4).

  1. There are some several points of concern related to the pollutant estimates used by the authors in the epidemiologic analysis. First, there is not a great deal of exposure variability for PM2.5 (IQR ~ 5 µg/m3) and authors should acknowledge this. Second and perhaps more importantly, is the low LUR model performance used to generate the PM2.5 exposure estimates. The authors acknowledge this in the discussion, however, it still seems hard for me to reconcile using a model with R2 ≈ 0.29 to generate exposure predictions for an environmental epi study. More attention should be devoted to these issues in the discussion of the observed results. Also, authors may wish to place their assumptions that typical/traditionally worse exposure measurement lead to bias to the null, in context with studies such as Szpiro et al., 2011; Sheppard et al., 2012; van Smeden et al., 2020; and others.

Response: The text was revised as suggested and a reference included. The following sentences was added to last paragraph of the Discussion section " Additionally, the less-than ideal exposure estimation in the study with low exposure variability for PM 2.5 (IQR~ 5 µg/m3) and a model with R2 ≈ 0.29 to generate exposure predictions could lead to bias to the null .35 Although data collected from several local sources such as housing and population density, waste burning, outdoor grilling, bus routes, and construction sites were found to be predictors of PM 2.5 concentrations, a lack of comprehensive data on such sources especially their transient nature could explain a significant part of the low variability found for PM 2.5.21“(Third last and second last sentence).

Round 2

Reviewer 3 Report

The authors have shown good effort in revising the manuscript according to the feedback they received. Here are a few additional points for them to consider:

- The same values from the original submission described as "Regression Coefficients" are presented in this revision as "Odds Ratios". Authors should be sure of what they are reporting and its interpretation. My understanding is that in a logistic regression the regression coefficients aren't necessarily the odds ratios.

- I disagree with reviewer 1 about figure 1. There is no need to make axes for NO2 and PM2.5 match - they are different pollutants. Also, I do not think the guidelines (indicating range on Y-axis?) are helpful. I'd suggest keeping the added A. and B. plot labels and Y-axis titles which weren't present on the original figure.

- Section 3.4 should be revised to clarify what measures are being presented (odds ratios, presumably).

- Tables 4 and 5: Authors need to indicate in the table captions what they are presenting (again, odds ratios and 95% CIs, presumably). Also, the tables share the same footnotes; should the 2-pollutant model say "NO2 and PM2.5" rather than "NO2 or PM2.5"?

- Authors have added a suggested note to the fact there is not a great deal of variability in estimated PM2.5 exposure, which I think is appropriate. An additional piece of information, in conjunction with the model’s poor performance, which may be relevant for the epi results, is that depending on the geographic scale of the neighborhoods/study region, low PM2.5 variability isn't necessarily unexpected (given it is generally considered a regional pollutant with less small-scale variability). A bit of geographic context may help here (e.g. size of study region, map, etc.).

Author Response

Response to the reviewer’s comments

  1. The same values from the original submission described as "Regression Coefficients" are presented in this revision as "Odds Ratios". Authors should be sure of what they are reporting and its interpretation. My understanding is that in a logistic regression the regression coefficients aren't necessarily the odds ratios

Response: Odds ratios were mistakenly reported as regression coefficients in the original submission but this was corrected in the revised manuscript. This is further clarified in this submission as indicated below in response to other reviewer’s comments.

  1. I disagree with reviewer 1 about figure 1. There is no need to make axes for NO2 and PM2.5 match - they are different pollutants. Also, I do not think the guidelines (indicating range on Y-axis?) are helpful. I'd suggest keeping the added A. and B. plot labels and Y-axis titles which weren't present on the original figure

Response: As suggested by the reviewer, the original figure 1 was included in this submission but with the A and B plot labels and Y axis titles included in figures

  1. Section 3.4 should be revised to clarify what measures are being presented (odds ratios, presumably)

Response: Section 3.4 was revised to clarify that odds ratios are being reported on.

  1. Tables 4 and 5: Authors need to indicate in the table captions what they are presenting (again, odds ratios and 95% CIs, presumably). Also, the tables share the same footnotes; should the 2-pollutant model say "NO2 and PM2.5" rather than "NO2 or PM2.5"?

Response: The captions of table 4 and 5 was revised to indicate that odds ratios and 95% CIs are being reported on. The footnote of Table 5 was revised to state “NO2 and PM2.5”

  1. Authors have added a suggested note to the fact there is not a great deal of variability in estimated PM2.5 exposure, which I think is appropriate. An additional piece of information, in conjunction with the model’s poor performance, which may be relevant for the epi results, is that depending on the geographic scale of the neighborhoods/study region, low PM2.5 variability isn't necessarily unexpected (given it is generally considered a regional pollutant with less small-scale variability). A bit of geographic context may help here (e.g. size of study region, map, etc.).

Response: The following sentence was added to the Discussion section” The low PM2.5 variability in the study in not necessarily unexpected considering the small scale of the four study neighbourhoods (< 40 km2) and the relatively small distances between the four sites (<500 Km). PM2.5 mostly reflects background pollution with low small-scale variability.”